

# Measurement of horizontal wind profiles in the polar stratosphere and mesosphere using ground based observations of ozone and carbon monoxide lines in the 230–250 GHz region: Proof of concept

D. A. Newnham[1], G. P. Ford[1,*], T. Moffat Griffin[1], and H. C. Pumphrey[2]

[1]British Antarctic Survey, High Cross, Madingley Road, Cambridge, CB3 0ET, United Kingdom
[2]School of GeoSciences, University of Edinburgh, Edinburgh, EH9 3FF, United Kingdom
[*]Now at Met Office, FitzRoy Road, Exeter, Devon, EX1 3PB, United Kingdom

*Correspondence to*: D. A. Newnham (dawn@bas.ac.uk)

**Abstract.** Meteorological and atmospheric models are being extended up to 80 km altitude but there are very few observing techniques that can measure stratospheric-mesospheric winds at altitudes between 20 km and 80 km to verify model data-sets. Here we demonstrate the feasibility of horizontal wind profile measurements using ground-based passive millimetre-wave spectroradiometric observations of ozone lines centred at 231.28 GHz, 249.79 GHz, and 249.96 GHz. Vertical profiles of horizontal winds are retrieved from forward and inverse modelling simulations of the line-of-sight Doppler-shifted atmospheric emission lines above Halley station (75°37'S, 26°14'W), Antarctica. For a radiometer with a system temperature of 1400 K and 30 kHz spectral resolution observing the ozone 231.28 GHz line we estimate that 12hr zonal and meridional wind profiles could be determined over the altitude range 25–74 km in winter, and 28–66 km in summer. Height-dependent measurement uncertainties are in the range 3–8 m s$^{-1}$ and vertical resolution ~8–16 km. Under optimum observing conditions at Halley a temporal resolution of 1.5hrs for measuring either zonal or meridional winds is possible, reducing to 0.5hr for a radiometer with a 700 K system temperature. Combining observations of the 231.28 GHz ozone line and the 230.54 GHz carbon monoxide line gives additional altitude coverage at 85±12 km. The effects of clear-sky seasonal mean winter/summer conditions, zenith angle of the received atmospheric emission, and spectrometer frequency resolution on the altitude coverage, measurement uncertainty, and height and time resolution of the retrieved wind profiles have been determined.

## 1 Introduction

The measurement of horizontal wind profiles in the 20–70 km altitude range remains a challenge for atmospheric science. Although a variety of in situ and remote sensing techniques, summarised in Fig. 1, are available for observing winds in the troposphere, lower stratosphere, and lower thermosphere the routine measurement of upper stratospheric and mesospheric winds have been difficult to obtain.

Ground-based radar systems such as meteor radars (Maekawa et al., 1993; Jacobi et al., 2007; Hoffmann et al., 2007), incoherent scatter radars (Alcaydé and Fontanari, 1986; Nicolls et al., 2010) and medium frequency radars (Briggs, 1980;



Hoffmann et al., 2007) profile winds down to 70 km. Mesosphere-Stratosphere-Troposphere (MST) radar techniques (e.g., Hooper et al., 2008; Lee et al., 2014) provide additional coverage over 0–20 km. Under clear-sky conditions the Rayleigh / Mie / Raman (RMR) LiDAR (Baumgarten, 2010; Hildebrand et al., 2012) measures winds throughout the middle atmosphere above Andenes, northern Norway but requires complex laser transmission and detection equipment and is not portable.

Ground-based Doppler LiDARs measure tropospheric and lower stratospheric winds (Gentry et al., 2000), with possible extension up to 50 km (Souprayen et al., 1999) and sodium LiDARs cover the altitude range 85–100 km (Williams et al., 2004). The ground-based E-region wind interferometer (ERWIN) measured upper atmospheric winds at three altitudes near the mesopause using airglow emissions from electronically-excited atomic oxygen $O(^1S)$, OH, and molecular oxygen $(O_2)$ (Gault et al., 1996a). Balloon-borne radiosondes provide detailed wind measurements in the troposphere and lower

stratosphere (e.g., Hibbins et al, 2005). Rockets can deploy falling targets tracked by radar (Müllemann and Lübken, 2005), or chemical tracers (Chu et al., 2007), but such campaigns are expensive and unsuitable for long-term monitoring. Infrasound measurements are used for investigating wind velocity variations in the stratosphere, mesosphere, and lower thermosphere (e.g., Le Pichon et al., 2010; Assink et al., 2012, 2013; Chunchuzov et al., 2015).

Over a 14 year period the space–borne High Resolution Doppler Imager (HRDI) (Burrage et al., 1996) measured wind

velocities typically between 60°N and 60°S and down to 50 km and contributed to the Upper Atmosphere Research Satellite (UARS) Reference Atmosphere Project (URAP) (Swinbank and Ortland, 2003) wind climatology. Also on UARS, the limb-sounding WIND Imaging Interferometer (WINDII) observed $O(^1S)$ airglow emission to measure winds over 90–110 km (Gault et al., 1996b). Since 2002 the TIMED Doppler Interferometer (TIDI) has been making limb–scanning measurements of $O(^1S)$ and $O_2$ (0–0) band emissions for determining horizontal winds above 60 km (Killeen et al., 2006). Satellite

measurements of mesospheric winds down to 80 km were determined from Aura Microwave Limb Sounder (MLS) observations of Doppler-shifted, Zeeman-split $O_2$ $\sigma^-$ and $\sigma^+$ lines around 118.75 GHz (Wu et al., 2008). Winds at 35–80 km were observed for a limited time over the latitude range 30°S to 55°N using ozone $(O_3)$ and hydrogen chloride emission line measurements in the 625–650 GHz range by the Sub-Millimetre-wave Limb-Emission Sounder (SMILES) (Baron et al., 2013). The Atmospheric Laser Doppler Instrument (ALADIN) on the forthcoming Atmospheric Dynamics Mission (ADM) Aeolus

satellite will make UV LiDAR measurements of winds in the troposphere and lower stratosphere (Stoffelen et al., 2005). The aim of the planned Stratospheric Wind Interferometer for Transport studies (SWIFT) is to make global measurements of middle stratospheric winds between 15–50 km using 8 μm $O_3$ emission lines (McDade et al., 2001; Shepherd et al., 2001).

For the reasons outlined above, global models such as the semi-empirical Horizontal Wind Model (HWM) (Drob et al., 2008; Drob et al., 2015) and reanalysis data-sets such as the European Centre for Medium-Range Weather Forecasts (ECMWF)

operational analyses (Dee et al., 2011) and NASAs Modern-Era Retrospective Analysis for Research and Applications (MERRA) (Rienecker et al., 2011) contain sparse information based on observations above 20 km, in contrast to the widespread operational data for the lower atmosphere. Systematic comparisons between co-located ground-based wind radiometer, LiDAR, and infrasound observations and re-analysis data found both temperature and horizontal wind speeds





deviate increasingly above 40 km as the assimilated observations became sparser (Le Pichon et al., 2015). Between altitudes of 40 km to 60 km the standard deviation of the mean difference in zonal winds exceeds 20 m s$^{-1}$ with the largest differences in winter when variability associated with large-scale planetary waves dominates. High-latitude mesospheric horizontal winds in HWM-07 (Drob et al., 2008) have standard deviations greater than 25 m s$^{-1}$ and significant differences with observed zonal

and meridional winds above 80 km (Sandford et al., 2010). With ECMWF producing specifications based on temperature soundings up to 75 km and fully assimilative high-top global circulation models being extended into the thermosphere, there is considerable urgency to verify the mean horizontal winds in models and meteorological (re)analyses. Superimposed on the mean flow are atmospheric tides and vertically-propagating waves that influence the planetary-scale circulation of the stratosphere, drive mesospheric circulation, and drive chemical transport and temperature gradients through pole-pole

meridional circulation, turbulent mixing, and diffusion (Brasseur and Solomon, 2005; Holton, 2004). These impact on the stability of the northern and southern hemisphere polar vortices where abrupt changes or even reversals of strong zonal winds during sudden stratospheric warming (SSW) events lead to strong dynamical and chemical coupling between the lower and upper atmosphere (Manney et al., 2008). New wind observing techniques for the Polar Regions are essential to better understand and parameterise these processes in circulation models for climate studies and numerical weather prediction.

Attention has recently turned to using ground-based microwave and sub-millimetre Doppler spectroradiometry to provide measurements of horizontal wind profiles covering the altitude range 30–85 km. By measuring the Doppler frequency shift of passive emission due to a rotational transition by a selected atmospheric molecule the line-of-sight wind speed can be determined (Clancy and Muhleman, 1993). The magnitude of the shift $\Delta \nu$ (in units of Hz) depends both on the wind speed along the line-of-sight (LOS) of the detector, $v_{LOS}$ (in m s$^{-1}$), and the rest frame frequency of the line, $\nu_0$ (in Hz):

$$\Delta \nu = \frac{v_{LOS}}{c} \nu_0 \qquad (1)$$

where c is the speed of light (in m s$^{-1}$). The horizontal wind speed (in m s$^{-1}$) is given by:

$$v_{wind} = v_{LOS} \sec \varepsilon \qquad (2)$$

where $\varepsilon$ is the zenith angle of the ground-based observation. By taking measurements of the Doppler shifts from opposite azimuthal directions the zonal and meridional wind components are determined. The frequency shifts are small, in the range

5 kHz to 200 kHz, but can be measured precisely using stable, high-resolution digital fast Fourier transform spectrometers (Klein et al., 2012). Wind speeds at 70–85 km above the South Pole were determined with uncertainty ±5 m s$^{-1}$ by observing the carbon monoxide (CO) line at 461 GHz (Burrows et al., 2007). Rüfenacht et al. (2012, 2014) used observations of the O$_3$ microwave emission line at 142.18 GHz by the ground-based wind radiometer (WIRA), and optimal estimation retrieval, to measure daily mean zonal and meridional wind profiles. Comparison of their measured time series from four locations at

polar, mid-, and tropical latitudes with ECMWF model data showed agreement to within 10% for meridional and zonal stratospheric winds, but significant differences of up 50% in mesospheric zonal wind speeds (Rüfenacht et al., 2014).





The aim of this work is to investigate the potential for retrieving wind speeds from ground-based 230–250 GHz radiometer observations of the polar atmosphere. Potential advantages of radiometry at these intermediate frequencies, in-between the microwave and sub-millimetre, include larger Doppler shifts than in the microwave, lower atmospheric opacity and signal attenuation and smaller Doppler line widths than in the sub-millimetre, and numerous isolated or overlapping $O_3$ emission

lines and a CO line centred at 230.54 GHz. The $O_3$ 231.28 GHz line intensity ($4.846 \times 10^{-23}$ cm molecule$^{-1}$ at 296 K) is over twice that for the 142.18 GHz line ($2.346 \times 10^{-23}$ cm molecule$^{-1}$ at 296 K), producing a more intense emission signal at source. We perform calculations for simulated wind retrievals above Halley station, Antarctica. Studies of mean winds, gravity waves, planetary waves, and atmospheric tides have utilised radiosonde balloon, imaging Doppler interferometer, and SuperDARN radar observations from Halley (Hibbins et al., 2006; Hibbins et al., 2009; Nielsen et al., 2009). These observations have been

limited to the troposphere, lower stratosphere, and upper mesosphere; co-located radiometric measurements could provide complementary observations filling the gap in altitudes at 25–75 km. In Sect. 2 the methodology for simulating millimetre-wave atmospheric spectra and performing wind retrievals is described. In Sect. 3 we present the wind retrieval results for different atmospheric conditions and radiometer instrument configurations, and the main conclusions are summarised in Sect. 4.

## 2 Methodology

### 2.1 Ground-based radiometer instrument and location

Atmospheric observations are simulated for a ground-based millimetre-wave radiometer located at Halley station (75°37'S, 26°14'W, 43 m above mean sea level), Antarctica. The instrument characteristics and performance are based on an existing total power radiometer (Espy et al., 2006; Newnham et al., 2011; Straub et al., 2013; Daae et al., 2014), a relatively portable

and robust instrument that has been deployed in the Polar Regions for semi-autonomous, continuous year-round operation. This instrument utilises a superconductor-insulator-superconductor (SIS) mixer at 4 K and low-noise amplifiers for sensitive heterodyne measurements of atmospheric spectra in the frequency region 230–250 GHz. Measurements by such a system achieve a single-sideband noise temperature of 200 K and a total system temperature of 1400 K. We also assess how wind measurements would be improved by using radiometers with system temperatures of 700 K and 1000 K. Spectral analysis of

the down-converted signal over a 300 MHz bandwidth, centred on the emission line(s) of interest, and with channel widths ≥10 kHz is based on the specifications of commercially-available chirp transform and fast Fourier transform (FFT) spectrometers for radiofrequency (RF) analysis. The effects of frequency errors arising from reference oscillator instabilities and spectrum baseline artefacts such as standing waves, due to interfering reflections of millimetre-wave or RF signals within the radiometer, are neglected as their effects have been shown to be small for similar wind measurements (Rufenacht et al.,

2014).

The radiometer is assumed to make total power measurements in turn of two calibration targets, one at ambient temperature and a cold reference load, and upward-viewing sky measurements at either two or four cardinal azimuthal directions and at a





fixed zenith angle. If the duration of each measurement is ~10 s, allowing for receiver mirror repositioning in each viewing direction, then one complete measurement cycle taking ~60 s would provide calibrated spectra in two azimuthally-opposite directions, i.e. east and west for zonal wind and north and south for meridional wind determination. Integration of measured data over repeated calibration cycles reduces the signal-to-noise of the calibrated atmospheric spectra.

The zenith angle of the sky observation affects the wind measurement in a number of ways. At higher zenith viewing angles the LOS Doppler shift due to the horizontal movement of the observed air mass is increased. The atmospheric column of ozone or carbon monoxide molecules increases with zenith angle, with angles of 45°, 60°, and 80° yielding geometric air mass factors of 1.4, 2.0, and 5.8 respectively. Atmospheric opacity will affect the wind measurement and may dominate at higher zenith angles under conditions of high precipitable water vapour (PWV).

At different slant angles the air mass observed, and its distance from the radiometer, will change. The maps in Fig. 2 show the location of Halley station and intercepts at altitudes of 25 km, 50 km, and 75 km for zenith angles of 45°, 60°, and 80°. At 45° the distances on the ground between Halley and the intercepts are 25 km, 50 km, and 75 km, neglecting refraction, whereas at 80° the distances are 142 km, 284 km, and 425 km. Since the zonal and meridional wind retrievals are derived by combining radiometric measurements at opposing east-west or north-south directions the choice of zenith angle will affect the spatial

resolution of wind structures and atmospheric dynamical effects, and inter-comparison with radiosonde, radar, and uniformly-gridded reanalysis data. Halley is typically inside the Antarctic polar vortex which can extend to 60°S (Turunen et al., 2009) meaning that winter-time observations, even at a zenith angle of 80°, would usually be well within the vortex edge. Halley station is on the Brunt Ice Shelf, a relatively flat location that would allow unobscured clear-sky views from the ground in all directions and at zenith angles reaching 80° or higher. Fig. 3 shows histograms of six-hourly zonal and meridional wind

data from ECMWF Interim re-analysis data (Dee et al., 2011) over the pressure (altitude) range 0.1–25 hPa (~64–25 km). The data are for 75.5°S, 26.5°W, the grid point closest to Halley, and cover the austral winter (June, July, August – JJA) and summer (December, January, February – DJF) periods over the 5yr period 2009–2014. The zonal winds in the middle atmosphere are predominantly westward in summer with mean speed -9 m s$^{-1}$ and more strongly eastward in winter with mean value +34 m s$^{-1}$. However the re-analysis data show a large range of zonal wind speeds, in particular during winter where

wind speeds span -36 m s$^{-1}$ to +126 m s$^{-1}$. The meridional winds are lighter with mean values of +1 m s$^{-1}$ in both winter and summer. The winter meridional wind speeds show a large range from -92 m s$^{-1}$ to +75 m s$^{-1}$.

## 2.2 Atmospheric spectra

The clear-sky atmospheric spectrum in the 230–250 GHz region above Halley is dominated by discrete $O_3$, $O_2$, and CO lines together with smoothly-varying continua due to water vapour, oxygen, and nitrogen. Weak emission lines in this frequency

range due to trace species such as nitric oxide and nitrogen dioxide are not included in our wind retrieval calculations. Winter (June, July, August - JJA) and summer (December, January, February - DJF) seasonal-average profiles of $O_3$, CO, water vapour, $O_2$, and temperature over the altitude range 0–120 km are calculated from 12 years of simulations by the Whole



Atmosphere Community Climate Model with Specified Dynamics (SD-WACCM) version 3.5.48 (Garcia et al., 2007; Marsh, 2011; Lamarque et al., 2012). The profiles are averages of the SD-WACCM output at 74.8°S and 76.7°S, the gridded latitudes closest to Halley, and are shown in Fig. 4. The continuum contribution from molecular nitrogen uses standard sub-Arctic profiles. Compared to the summer case, winter-time $O_3$ VMR is higher in the secondary ozone layer centred at $10^{-3}$ hPa

(~96 km) and in the seasonal tertiary layer at ~0.05 hPa (~70–75 km). In summer $O_3$ VMR is higher in the upper stratosphere between 0.3–0.8 hPa (~54–38 km). Mesospheric CO VMR is higher in winter, due to strong descent in the southern polar vortex, but SD-WACCM may underestimate the seasonal CO variability observed by Aura MLS (Pumphrey et al., 2007). Higher summer mean temperatures and tropospheric water vapour VMR lead to increased PWV and atmospheric opacity at millimetre wavelengths. For Halley the mean PWV calculated from the SD-WACCM data is 6.58 mm in summer and 1.18 mm

in winter. The lower quartile PWV in winter is 0.92 mm, i.e. for 25% of the time during winter months (JJA) PWV is at this value or lower with a mean value of 0.76 mm.

The Atmospheric Radiative Transfer Simulator (ARTS) (version 2.2.0) available at http://www.radiativetransfer.org/ is the forward model used in this study (Buehler et al., 2005; Eriksson et al., 2011). ARTS is a line-by-line model that can simulate radiances from the infrared to the microwave, and has been validated against other models in the millimetre spectral range

(Melsheimer et al., 2005). It includes contributions from spectral lines and continua via a choice of user-specified parameterisations. For our work, we use the Planck formalism for calculating brightness temperatures and spectroscopic line parameters are taken from the HIgh resolution TRANsmission (HITRAN) molecular database 2012 (Rothman et al., 2013). The oxygen continuum according to Rosenkranz (1998), nitrogen self-broadening (Liebe et al., 1993), and water vapour continuum (Rosenkranz, 1993) are included in the model. Survey clear-sky atmospheric spectra covering the 228 GHz to

252 GHz range, calculated on a 10 MHz frequency grid, are shown in Fig. 5a. The higher baseline brightness temperatures, and reduced emission line signals in summer at Halley are due primarily to increased atmospheric opacity at higher tropospheric temperature and water vapour VMR. The spectra show the most intense $O_3$ emission lines, the $J = 2 \rightarrow 1$ CO line centred at 230.54 GHz, and a $^{16}O^{18}O$ line centred at 233.95 GHz. The nearly constant mixing ratio of $^{16}O^{18}O$ could make the 233.95 GHz emission line suitable for profiling winds throughout the stratosphere and mesosphere. However, the

Zeeman-splitting of the $^{16}O^{18}O$ line would need to be accurately modelled in the forward model and retrieval algorithms (Navas-Guzmán et al., 2015) and such analysis is not included in this report. The enlarged 300 MHz-wide plots shown in Figs. 5b–d show the target frequencies for wind retrievals with the molecular line-shapes dominated by contributions from Doppler- and pressure broadening.

For wind retrievals, Doppler-shifted atmospheric spectra are calculated for ground-based north- and south-, or east- and west-,

pointing azimuthal directions with zonal and meridional winds at fixed values of +20 m s$^{-1}$ at all altitudes from the ground to 120 km. The frequency spacing of the atmospheric spectra is 10 kHz within 2.3 MHz of line centres, 100 kHz at 2.3–9.4 MHz from the line centres, and 1 MHz beyond 9.4 MHz from the line centres. This variable frequency grid ensures the spectral features are accurately represented while reducing the computing resource needed for wind retrieval calculations.





The statistical fluctuation $\Delta T$ (K) in the total system temperature $T_{sys}$ (K) is calculated according to the ideal radiometer equation (Kraus, 1966):

$$\Delta T = \frac{T_{sys}}{\sqrt{t \Delta f}} \tag{3}$$

where $t$ is observation time (in s) and $\Delta f$ is the frequency resolution (in Hz) of the radiometer.

## 2.3 Wind retrieval

The simulated atmospheric spectra are inverted into altitude profiles of zonal and meridional wind speed using an iterative optimal estimation method (OEM) (Rodgers, 2004) implemented in the Qpack (a part of atmlab v2.2.0) software package (Eriksson et al., 2005). A detailed description of wind profile retrievals using ARTS and Qpack is given by Rufenacht et al. (2014). Here we focus on the description of our specific retrieval setups and discussion of results for wind speed estimations from simulated 230–250 GHz measurements at Halley, Antarctica. Iterative absorption calculations in ARTS are performed line-by-line inside the radiative transfer calculation, rather than using pre-calculated look-up tables, in order to accurately model atmospheric spectra at the Doppler shifted frequencies (Buehler et al., 2011).

$O_3$ or CO VMR profiles and zonal / meridional wind profiles are retrieved at altitude levels 0–120 km with a 1 km spacing, where hydrostatic equilibrium is assumed for the altitude and pressure. The a priori wind speed is 0 m s$^{-1}$ for all altitudes, with diagonal elements of the covariance set according to a wind speed uncertainty of 20 m s$^{-1}$ based on the ECMWF reanalysis data (Fig. 3). Rufenacht et al. (2014) showed that the retrieved wind profiles are relatively insensitive and unbiased to different a priori wind profiles even when the a priori assumption is far from the true wind. The shape of the covariance is set to decrease linearly towards the off-diagonal elements with correlation length adjusted to match the altitude resolution of initial retrievals using only the diagonal covariance matrix elements. The correlation length is typically in the range 0.6–0.8 of a pressure decade (approximately 10–12 km).

The $O_3$ and CO a priori VMR profiles used in wind retrievals are those used to calculate the simulated atmospheric spectra, apart from calculations to test the effect of scaling the original VMR profiles at all altitudes by 80%, 90%, 110%, and 120%. The diagonal elements in the covariance of the $O_3$ and CO a priori are fixed at the square of 50% of the VMR values. The shape of the covariance is set to linearly decrease towards the off-diagonal elements with a correlation length of a fifth of a pressure decade (approximately 3 km).

Nominal wind retrievals were performed for simulated clear-sky 12hr observations of the $O_3$ 231.28 GHz line at 60° zenith angle from Halley in mean winter and summer conditions and for a radiometer with 1400 K system temperature and 30 kHz frequency resolution. The averaging kernels (AVKs) for every sixth retrieved altitude are shown in Fig. 6a and Fig. 6e for the winter (JJA) and summer (DJF) cases respectively. The AVKs describe the relationship between the true, a priori, and retrieved atmospheric states (Rodgers, 2004). None of the AVKs peak at pressure levels above 0.02 hPa (~74 km) in winter, or above 0.08 hPa (~66 km) in summer due to the combination of Doppler broadening dominating over pressure broadening and low $O_3$ VMR in particular during summer when the seasonal tertiary ozone layer at 70–75 km is not present. The lowest AVK





peaks are at 27 hPa (~25 km) in winter and 17 hPa (~28 km) in summer. The retrievals in summer are also adversely affected by higher PWV and atmospheric opacity.

The sum of the AVKs at each altitude, called the measurement (or total) response (MR), represents the extent to which the measurement contributes to the retrieval solution as compared to the amount of influence of the a priori at that altitude

(Christensen and Eriksson, 2013). The altitude range where the retrieved wind profile has a high degree of independence from the a priori is estimated by MR values higher than 0.8. The retrieval range is shown by the thicker sections of the black lines in Fig. 6a and Fig. 6e and is 0.02–27 hPa (~74–25 km) for mean winter-time conditions and 0.08–17 hPa (~66–28 km) in summer. Outside of these altitudes (i.e. below 25 km and above 74 km in winter, and below 28 km and above 66 km in summer) the MR weakens and wind values in these regions should be interpreted with caution as the information from the a

priori becomes important. The AVKs indicate the range of altitudes over which the retrieved wind speeds has smoothed the information in the data. Thus, the full-width half-maximum (FWHM) width of the kernels provide a measure of the vertical resolution of the retrieved profile. The FWHM values shown in Fig. 6b and Fig. 6f indicate altitude resolutions of 10.0–15.5 km and 9.5–14.9 km over the winter and summer retrieval ranges respectively, similar to the WIRA instrument performance (Rüfenacht et al., 2014). The OEM calculations provide observation errors ($\sigma_{obs}$) and total retrieval (observation

plus smoothing) errors ($\sigma_{tot}$) to give further diagnostic estimates of the uncertainty of retrieved profiles. The observation errors describe how the retrieved profiles are affected by measurement noise and are shown in Fig. 6c and Fig. 6g, with typical values of 4.8 m s$^{-1}$ in winter and 6.1 m s$^{-1}$ in summer. The observation errors are small outside of the range of the AVK peaks as the retrieval tends to the a priori values in these regions and the contribution from the measurement is small. The total retrieval errors shown in Fig. 6d and Fig. 6h are in the range 7.8–15.9 m s$^{-1}$ in winter and 9.8–15.3 m s$^{-1}$ in summer, and outside

the range of AVK peaks tend towards the a priori standard deviation of 20 m s$^{-1}$.

We have also assessed measurement uncertainties using Monte Carlo simulations, as was done by Rüfenacht et al. (2014) in their wind retrievals using the $O_3$ 142.18 GHz emission line. Our Monte Carlo error analysis results, using 500 repeat zonal wind retrievals to test the retrieval algorithm's ability to reproduce the "true" state of the atmosphere, are shown in Fig. 7. Fig. 7c and Fig. 7h are for the winter and summer nominal cases, i.e. simulated clear-sky, 12hr observations (i.e. 6hrs in east

and 6hrs in west directions) of the $O_3$ 231.28 GHz line using a ground-based radiometer with a 1400 K system temperature, 30 kHz frequency resolution, and 60° zenith viewing angle. Over the trustable altitude range the mean wind profile is 20.3 m s$^{-1}$ in winter and 19.6 m s$^{-1}$ in summer, both values within 2% of the "true" value of 20.0 m s$^{-1}$. The standard deviation of the individual retrievals is an estimator for the uncertainty of the wind retrieval, and the mean values of 4.8 m s$^{-1}$ in winter and 6.1 m s$^{-1}$ in summer match the mean observation errors determined from single retrievals. This is not surprising as both

parameters are dependent on the signal-to-noise ratio of the input spectrum. Our calculated observation error is considerably smaller than the 12–20 m s$^{-1}$ range reported for the WIRA new single sideband receiver (Rüfenacht et al., 2014). The improvement is probably largely due to the low noise levels for a SIS mixer receiver used in our spectrum simulations, and the larger Doppler shifts and higher line intensity of $O_3$ at 231.28 GHz compared to the 142.18 GHz line. However it should be noted that the location and atmospheric conditions for the calculations differ and an exact comparison between the actual,



or likely, performance of each instrument cannot be made using these data. Figs. 7a–b, d–e and Figs. 7f–g, i–j show the Monte Carlo simulation results for winter and summer conditions respectively when the a priori $O_3$ VMR profiles are scaled by 80%, 90%, 110%, and 120% of the "true" profiles used to simulate the atmospheric spectra. These calculations indicate that 20% uncertainties in a priori $O_3$ VMR could introduce biases in the retrieved wind profiles of 15–21%. However, careful

construction of a priori data-sets and configuring of retrieval parameters may mitigate against such errors. Uncertainties in spectroscopic parameters and temperature profiles are not expected to cause significant biases in the retrieved winds, as was reported by Rüfenacht et al. (2014).

## 3 Results

In the following sections horizontal wind retrieval results are presented for simulated scenarios where five instrument

parameters are varied: - the zenith viewing angle of the ground-based observation, the instrument's frequency resolution, the radiometer system temperature, the observed emission line, and the measurement time. The main results are summarised in Table 1. The wind retrievals shown are for the zonal component of the horizontal wind and quoted measurement times are for corresponding observations made in opposing east and west directions. Identical results are obtained for meridional wind analyses where the simulated measurements are in north and south pointing directions.

### 3.1 Effect of zenith angle

In order to observe a Doppler shift in the millimetre-wave atmospheric emission arising from the horizontal motion of the molecules in the air mass, observations must be made at non-zero zenith angles. The results of wind retrievals for simulated observations with zenith angles of 45°, 60°, and 80° are shown in Fig. 8 and in Table 1. A zenith angle of 80° gives the best results for the Halley mean winter-time conditions considered, with winds retrieved over 0.02–38.8 hPa (~74–23 km).

Summer-time coverage at 80° is over a narrower pressure range, 0.12–16.9 hPa (~63–28 km), and at a 60° angle the retrieval for summer conditions reaches slightly higher, covering 0.08–16.9 hPa (~66–28 km). The 80° observations give the best height resolutions in winter, estimated at 9–15 km from the AVK FWHMs, and smallest observation errors, in the range 3.3–7.2 m s$^{-1}$. In summer both the height resolutions and observation errors at 60° are slightly smaller than for the corresponding 45° or 80° summer retrievals over most of the altitudes where wind information is retrieved from the measurement. With 30°

zenith angle simulations the retrieval total response did not exceed 0.8 at any altitude. The seasonal variability of the wind retrieval quality indicates that the wind information contained in the Doppler-shifted $O_3$ 231.28 GHz emission line signals depends on zenith angle-dependent factors including line-of-sight Doppler shifts, air mass factor, and atmospheric opacity as well as the effects of different seasonal $O_3$ VMR and temperature profiles. For the zenith angles considered, 80° provides the highest air mass factor and highest LOS Doppler shift with a relatively small 10° angle between the observing beam and the

horizontal plane giving high projection efficiency. The optimum zenith angle will vary with ground-based location and the

atmospheric conditions at the time of making observations, as well as considerations of the spatial distances between measurements in opposite directions. From a practical perspective, the range of zenith angles that the instrument can be pointed may be limited by the requirement to have a clear, unobscured view of the sky continuously throughout measurements in each different azimuthal direction.

## 3.2 Effect of instrument frequency resolution

The results of wind retrievals for simulated observations with instrument channel spacing of 10 kHz, 30 kHz, 100 kHz, 300 kHz, and 1 MHz are shown in Fig. 9 and Table 1. Increasing the frequency resolution from 10 kHz to 300 kHz has little effect on horizontal wind retrievals from 12hr simulated observations of the 231.28 GHz $O_3$ emission line in mean winter-time or summer-time conditions at Halley. Height resolutions, estimated from the AVK FWHMs, and observation errors are similar for all five resolutions below 1 hPa (~48 km). At 1 MHz resolution the upper limit where the measured data dominate the wind retrieval is substantially lower, at 0.12 hPa (~63 km) in winter and 0.25 hPa (~58 km) in summer and, close to this altitude limit, the observation errors increase to 7.7 m s$^{-1}$.

The lack of sensitivity of the simulated wind retrievals to frequency resolution, at least up to 300 kHz channel spacing, is surprising given that the LOS Doppler shift of the $O_3$ 231.28 GHz emission line is 13.4 kHz for a 20 m s$^{-1}$ horizontal wind viewed at 60° zenith angle. However, the Jacobian and gain matrices for the wind retrievals indicate the Doppler-shifted spectral response should be adequately sampled at instrument resolutions up to 300 kHz for altitudes where the $O_3$ measurement contributes significantly to the retrieval. Frequency resolution in the range 10–30 kHz is needed to determine upper mesospheric winds from CO 230.54 MHz observations where Doppler shifts above 75 km result in changes to the spectral distribution much closer to the emission line centre. The values of the Jacobian describing the $O_3$ wind retrieval, normalised by the layer thickness of the retrieval grid for observations at a 60° zenith angle, are shown in Figure 10 with typical values around 0.15 mK (m s$^{-1}$)$^{-1}$km$^{-1}$. The effect of wind variations of 20 m s$^{-1}$ on the measured atmospheric brightness temperature will therefore be small, of the order of 3 mK km$^{-1}$. For an ideal instrument the baseline signal-to-noise varies as $1/\sqrt{\Delta f}$ for a fixed measurement time, i.e. a factor 3.3 improvement for a ten-fold change in frequency resolution ($\Delta f$) from 30 kHz to 300 kHz. Small changes in brightness temperature arising from the Doppler-shift of emission lines may be better characterised by reduced frequency resolution measurements if the resulting higher signal-to-noise offsets the effect of reduced sampling of the frequency distribution of the emission.

## 3.3 Effect of radiometer system temperature

The results of zonal wind retrievals for simulated observations with radiometer system temperatures of 700 K, 1000 K, and 1400 K are shown in Fig. 11 and Table 1. The altitude range, height resolution, and observation errors of the wind retrievals all improve with lower system temperature, due to the higher signal-to-noise of the measurements. The signal-to-noise of the simulated measurements is based on real instrument data for a radiometer operating at 230–250 GHz and varies with $T_{sys}$, i.e.





the noise level halving when $T_{sys}$ changes from 1400 K to 700 K. This impacts directly on the measurement of small changes in atmospheric emission spectra arising from Doppler shift perturbations and the retrieval of wind information, as discussed previously in Section 3.2.

**3.4 Effect of measurement time**

The results of zonal wind retrievals for simulated 4hr, 6hr, and 12hr measurements (i.e. 2hr, 4hr, and 6hr observations in east and west directions) with a system temperature of 1400 K are shown in Fig. 12 and Table 1. For the 4hr summer-time observation, and measurement times below 4hrs (not shown) for the winter case, the retrieval total response does not exceed 0.8 at any altitude. The minimum measurement times for successfully retrieving zonal or meridional winds is 4hrs in mean winter-time conditions at Halley and 6hrs in summer for the specified instrument configuration. The altitude range, height

resolution, and observation error of the wind retrievals all improve as measurement time $t$ increases, due to higher signal-to-noise which varies as $\sqrt{t}$. The time resolution of horizontal wind measurements is limited by the need to acquire atmospheric spectra with sufficiently low noise for the wind retrieval to succeed. During optimum observing conditions at Halley the measurement time is considerably reduced, as shown in Fig. 13 and Table 1 for mean winter-time atmospheric profiles when PWV is in the lower quartile, i.e. during a quarter of winter (JJA) conditions at Halley when PWV is below

0.92 mm, which corresponds to 23 days per year. A minimum measurement time of 1.5hrs is achieved using a 1400 K system temperature radiometer, reducing to 0.5hr resolution for a 700 K radiometer. At lower measurement times the trade-off is a modest reduction in altitude coverage and resolution, and increased observation error.

**3.5 Choice of emission line(s)**

The results of zonal wind retrievals for simulated 12hr measurements of the $O_3$ 231.28 GHz, $O_3$ 250 GHz, CO 230.54 GHz,

and combined CO 230.54 GHz and $O_3$ 231.28 GHz emission lines, with a radiometer system temperature of 1400 K and 60° zenith viewing angle are shown in Fig. 14 and Table 1. Using either the single 231.28 GHz $O_3$ line or the pair of blended $O_3$ lines centred at 249.79 GHz and 249.96 GHz (see Fig. 2) the altitude range, height resolution, and observation errors of the retrievals are very similar for mean winter-time conditions. However in summer the retrieval using the pair of blended lines covers higher altitudes, reaching 0.05 hPa (~69 km) with smaller observation and total retrieval errors.

The retrieval using the CO 230.54 GHz emission line yields wind information over 0.0009–0.03 hPa (~97–73 km) in both summer and winter conditions. The retrieval altitudes correspond to the higher CO mixing ratios in the upper mesosphere and lower thermosphere (75–100 km) where CO is mainly produced by ultraviolet photo-dissociation of carbon dioxide ($CO_2$). At lower altitudes where CO mixing ratios are low, due primarily to the oxidation reaction with hydroxyl (OH) to form $CO_2$ (Minschwaner et al., 2010), the MR is below 0.8 and wind values in these regions should be interpreted with caution as the

information from the a priori becomes important. The CO wind retrieval corresponds to an altitude of 85±12 km where $O_3$ analysis alone does not reliably retrieve wind information. The height resolutions, estimated at ~20 km in winter and ~16 km



in summer from the AVK FWHMs, are coarser than for the $O_3$ wind retrievals whereas the observation errors are smaller at ~3 m s$^{-1}$ in both winter and summer conditions at Halley. Our simulated wind retrieval is comparable to the polar mesospheric wind speeds above the South Pole determined with uncertainty ±5 m s$^{-1}$ by observing the CO line at 461 GHz (Burrows et al., 2007).

5    Wind retrievals using the combined CO 230.54 GHz and $O_3$ 231.28 GHz observations provide the broadest coverage, 0.001–27 hPa (~96–25 km) in winter and 0.001–20 hPa (~96–27 km) in summer. Over this range the height resolutions, observation errors, and total errors are similar to those for the separate CO 230.54 GHz and $O_3$ 231.28 GHz retrievals, apart from at the overlapping altitudes (~75–80 km) in winter where the AVK FWHMs are as high as 32 km. Fine tuning the OEM wind retrieval for the combined analysis may optimise the information retrieval from the two emission lines and eliminate this artefact in the upper mesospheric data.

## 4 Conclusions

The proof-of-concept simulations demonstrate that polar middle atmosphere winds could be profiled with ~5 m s$^{-1}$ measurement uncertainty and ~12 km altitude resolution using ground-based passive millimetre-wave measurements of Doppler-shifted $O_3$ emission lines in the 230–250 GHz region and OEM retrieval. The effects of clear-sky seasonal mean winter/summer conditions, zenith angle of the received atmospheric emission, and spectrometer frequency resolution on the altitude coverage, measurement uncertainty, and height and time resolution of the retrieved wind profiles have been determined. We have used simulations of atmospheric spectra from Halley station, Antarctica as a test case. For a SIS-mixer based receiver the minimum observation times for determining zonal or meridional wind profiles would be 0.5–1.5hrs during optimal winter-time conditions, and more typically ~4hrs in winter and ~6hrs during summer. The millimetre-wave radiometry technique would allow stratospheric and mesospheric wind observations covering the altitude range 25–74 km in winter and 28–66 km in summer, with extension up to ~96 km by also observing the CO 230.54 GHz emission line. Such observations would complement established radiosonde and radar techniques that provide data on wind speeds in the troposphere and upper mesosphere. This would allow more detailed investigations of important dynamical and chemical transport processes associated with large-scale planetary wave activity and atmospheric tides that perturb the mean circulation patterns. A next stage would be to upgrade an existing 230–250 GHz radiometer, modifying the sky-viewing mirror at the front-end of the receiver to make automated observations at different azimuthal and zenith directions, and performing atmospheric measurements and retrievals to characterise the instrument performance and wind measurement capability under different conditions in the field.

30    *Acknowledgements.* This work has been supported by the UK's Natural Environment Research Council (NERC) Technologies Proof-of-Concept grant reference NE/L012197/1 awarded to DAN, TMG, and HCP. We acknowledge ECMWF for the re-analysis data. The authors thank the ARTS and Qpack development teams and P. Eriksson at Chalmers University of





Technology and P. Kirsch at BAS for assistance configuring and running the code, E. C. Turner at BAS for processing ERA data, D. Marsh at the National Center for Atmospheric Research (NCAR) for providing SD-WACCM data, and N. Kämpfer, R. Rüfenacht, and A. Murk at the University of Bern and N. J. Mitchell at the University of Bath for helpful discussions.

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



**Table 1.** Summary of wind retrieval results. Numbers in bold are for the nominal retrieval case in each parameter category.

| Parameter | Value | Pressure range / hPa Winter | Summer | Altitude range / km Winter | Summer | AVK FWHM / km Mean (range) Winter | Summer | $\sigma_{obs}$ / m s$^{-1}$ Mean (range) Winter | Summer | $\sigma_{tot}$ / m s$^{-1}$ Mean (range) Winter | Summer |
|---|---|---|---|---|---|---|---|---|---|---|---|
| Zenith angle | 45° | 0.03–16 | 0.10–11 | 29–73 | 31–64 | 13.2 (10.9–15.7) | 12.9 (10.2–15.4) | 5.2 (4.3–7.4) | 6.5 (5.5–7.7) | 9.9 (8.1–15.0) | 12.0 (10.2–14.9) |
| | **60°** | **0.02–27** | **0.08–17** | **25–74** | **28–66** | **12.1 (10.0–15.5)** | **12.2 (9.5–14.9)** | **4.8 (3.7–7.7)** | **6.1 (5.0–7.7)** | **9.9 (7.8–15.9)** | **11.7 (9.8–15.3)** |
| | 80° | 0.02–39 | 0.12–17 | 23–74 | 28–63 | 11.4 (9.3–14.7) | 12.6 (9.9–15.4) | 4.5 (3.3–7.2) | 6.5 (5.5–7.8) | 9.7 (7.5–16.1) | 12.3 (10.5–15.3) |
| $\Delta f$ | 10 kHz | 0.02–27 | 0.08–17 | 25–74 | 28–66 | 12.1 (10.0–15.5) | 12.2 (9.5–14.9) | 4.8 (3.7–7.7) | 6.1 (5.0–7.7) | 9.9 (7.8–15.9) | 11.7 (9.8–15.3) |
| | **30 kHz** | **0.02–27** | **0.08–17** | **25–74** | **28–66** | **12.1 (10.0–15.5)** | **12.2 (9.5–14.9)** | **4.8 (3.7–7.7)** | **6.1 (5.0–7.7)** | **9.9 (7.8–15.9)** | **11.7 (9.8–15.3)** |
| | 100 kHz | 0.02–27 | 0.08–17 | 25–74 | 28–66 | 12.1 (10.0–15.5) | 12.2 (9.5–14.9) | 4.8 (3.7–7.7) | 6.1 (5.0–7.7) | 9.9 (7.8–15.9) | 11.7 (9.8–15.3) |
| | 300 kHz | 0.02–27 | 0.09–17 | 25–74 | 28–65 | 12.4 (10.1–15.6) | 12.1 (9.5–14.9) | 4.7 (3.5–7.6) | 6.1 (5.0–7.7) | 9.6 (7.5–15.7) | 11.7 (9.8–15.1) |
| | 1 MHz | 0.12–23 | 0.25–17 | 26–63 | 28–58 | 12.4 (10.0–15.5) | 11.6 (9.6–13.7) | 4.9 (3.7–7.3) | 5.8 (4.8–7.7) | 9.7 (7.8–13.4) | 11.0 (9.4–13.8) |
| $T_{sys}$ | 700 K | 0.02–39 | 0.04–35 | 23–75 | 23–71 | 10.5 (8.5–13.8) | 11.3 (8.5–13.9) | 4.0 (3.0–7.0) | 4.9 (3.6–7.7) | 9.5 (7.3–16.7) | 10.0 (8.0–15.1) |
| | 1000 K | 0.02–32 | 0.06–23 | 24–74 | 26–68 | 11.3 (9.3–14.9) | 11.7 (9.0–14.4) | 4.4 (3.3–7.4) | 5.4 (4.2–7.5) | 9.6 (7.5–15.8) | 10.5 (8.6–14.9) |
| | **1400 K** | **0.02–27** | **0.08–17** | **25–74** | **28–66** | **12.1 (10.0–15.5)** | **12.2 (9.5–14.9)** | **4.8 (3.7–7.7)** | **6.1 (5.0–7.7)** | **9.9 (7.8–15.9)** | **11.7 (9.8–15.3)** |
| Measure-ment time | 4hrs | 0.03–13 | - | 30–73 | - | 13.5 (11.1–16.3) | - | 5.5 (4.6–7.3) | - | 10.3 (8.5–15.4) | - |
| | 6hrs | 0.03–16 | 0.01–11 | 29–73 | 31–64 | 12.9 (10.7–15.5) | 13.0 (10.3–15.5) | 5.2 (4.3–7.3) | 6.5 (5.6–7.7) | 10.1 (8.2–15.3) | 12.0 (10.3–15.1) |
| | **12hrs** | **0.02–27** | **0.08–17** | **25–74** | **28–66** | **12.1 (10.0–15.5)** | **12.2 (9.5–14.9)** | **4.8 (3.7–7.7)** | **6.1 (5.0–7.7)** | **9.9 (7.8–15.9)** | **11.7 (9.8–15.3)** |
| Emission line | **$O_3$ 231.28 GHz** | **0.02–27** | **0.08–17** | **25–74** | **28–66** | **12.1 (10.0–15.5)** | **12.2 (9.5–14.9)** | **4.8 (3.7–7.7)** | **6.1 (5.0–7.7)** | **9.9 (7.8–15.9)** | **11.7 (9.8–15.3)** |
| | $O_3$ 250 GHz | 0.02–27 | 0.05–20 | 25–75 | 27–69 | 11.1 (9.4–13.8) | 12.0 (9.4–14.8) | 4.3 (3.4–7.4) | 5.3 (4.0–7.4) | 9.7 (7.6–16.4) | 10.2 (8.4–14.7) |
| | CO 230.54 GHz | 0.0009–0.03 | 0.0009–0.03 | 73–97 | 73–97 | 19.6 (19.5–20.1) | 16.0 (15.5–16.4) | 2.7 (1.7–4.7) | 2.6 (2.3–3.4) | 12.2 (9.2–16.3) | (8.0–15.4) |
| | CO 230.54 GHz + $O_3$ 231.28 GHz | 0.001–27 | 0.001–20 | 25–96 | 27–96 | 14.8 (10.3–32.3) | 13.8 (10.4–16.8) | 4.0 (2.3–7.3) | 5.2 (2.4–7.7) | 10.2 (7.5–16.1) | (8.3–15.4) |





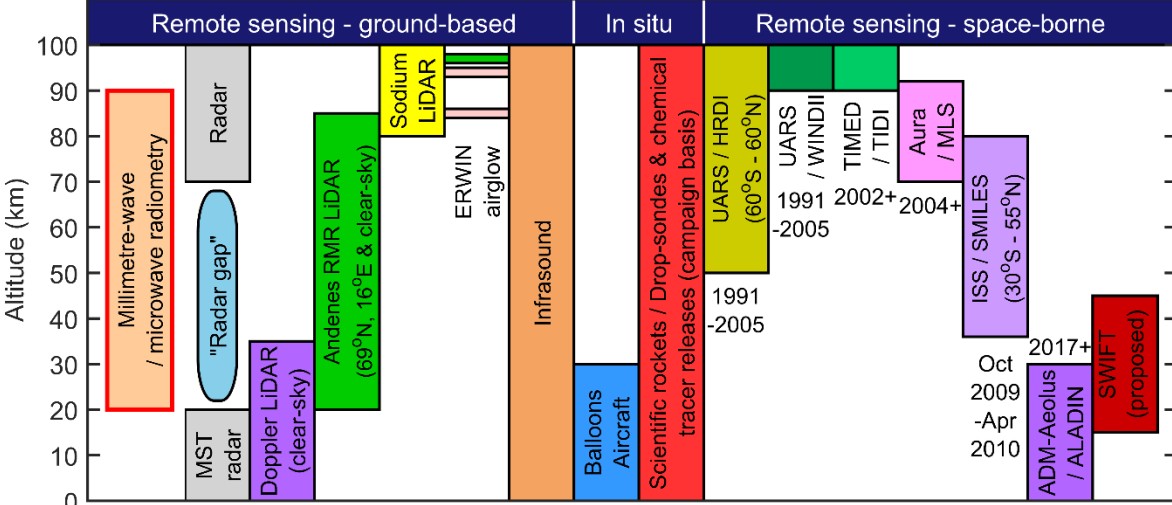

**Figure 1.** Overview of atmospheric wind measurement techniques with their altitude coverage. For space-borne sensors the abbreviations for the satellite / instrument names are given with years of operation.





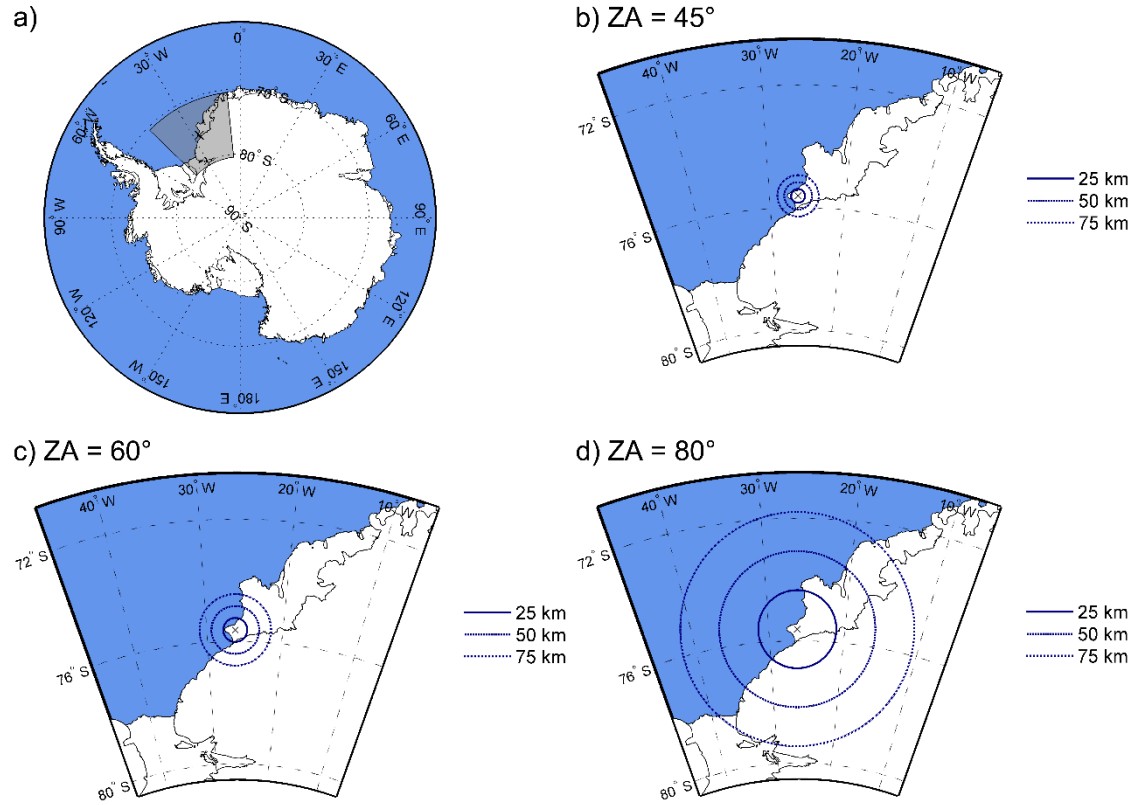

**Figure 2.** Maps showing a) the location of Halley station, Antarctica and the grey shaded region enlarged in b)–d) where the concentric circles are intercepts at altitudes of 25 km, 50 km, and 75 km for b) 45°, c) 60°, and d) 80° zenith viewing angle (ZA), neglecting refraction. '×' shows the location of Halley station (75°37'S, 26°14'W). Projection is azimuthal-polar stereographic for a) and Lambert conformal conic for b)–d). Coastline data are from Global Self-consistent, Hierarchical, High-resolution Geography Database (GSHHG) version 2.3.4.





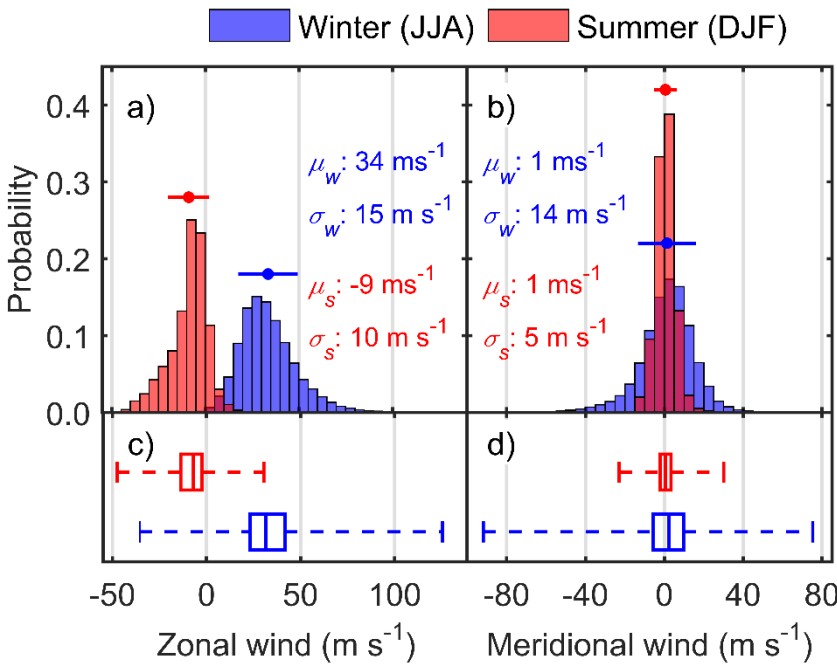

**Figure 3.** Histograms of a) zonal and b) meridional winds over the pressure range 0.1–25 hPa (~64–25 km) in winter (JJA, blue) and summer (DJF, red) from 6-hourly ERA-interim data for 2009–2014 at grid-point 75.5°S, 26.5°W. The horizontal error bars show the mean winds ($\mu_w$, $\mu_s$) and standard deviations ($\sigma_w$, $\sigma_s$) for winter (JJA) and summer (DJF) respectively. The corresponding boxplots of the c) zonal and d) meridional wind data indicate the minima, lower quartile (25[th] percentile), median, upper quartile (75[th] percentile), and maxima wind speeds. +ve zonal winds are eastwards and +ve meridional winds are northwards.





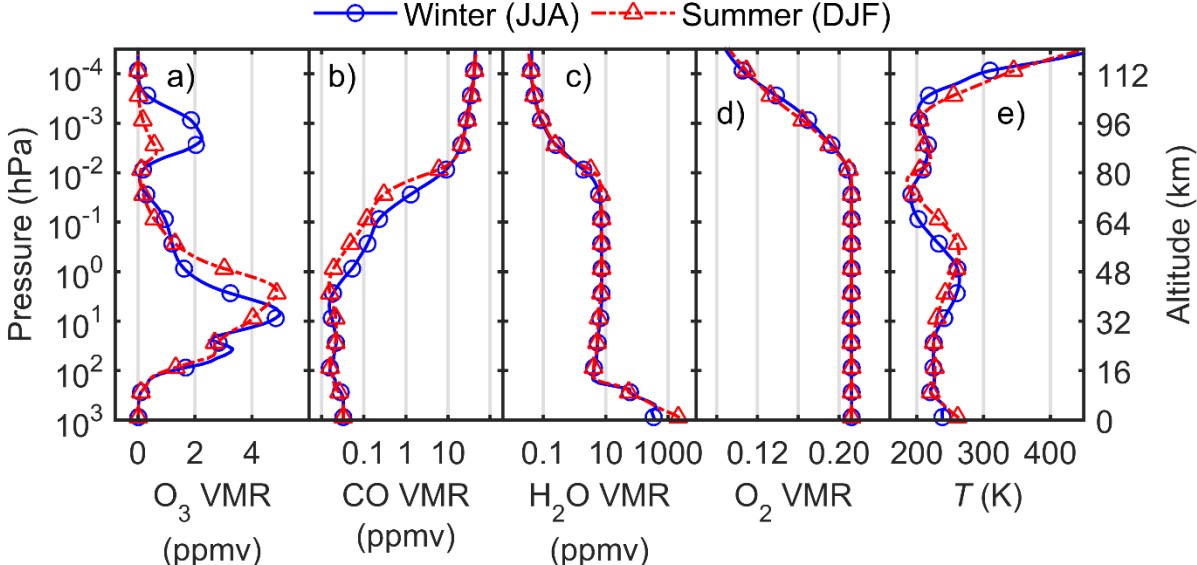

**Figure 4.** Seasonal mean winter (JJA, solid blue line) and summer (DJF, dashed red line) atmospheric profiles for a) $O_3$ VMR, b) CO VMR, c) water vapour VMR, d) $O_2$ VMR, and e) temperature from SD-WACCM simulation data.





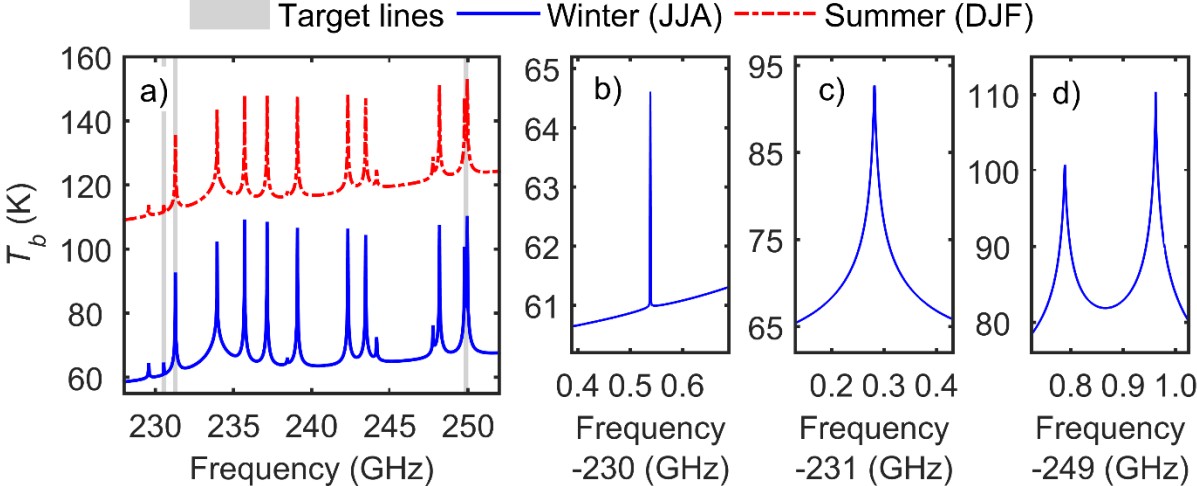

**Figure 5.** Simulated ground-based atmospheric brightness temperature spectra for clear-sky, 60° zenith angle viewing conditions at Halley station (75°37'S, 26°14'W), Antarctica for a) the frequency range 228–252 GHz in winter (JJA, solid blue line) and summer (DJF, dashed red line). The 300 MHz wide target frequencies, highlighted as grey shaded panels in a), are shown enlarged for the b) 230.54 GHz CO, c) 231.28 GHz $O_3$, and d) 249.79 GHz and 249.96 GHz $O_3$ lines in winter.





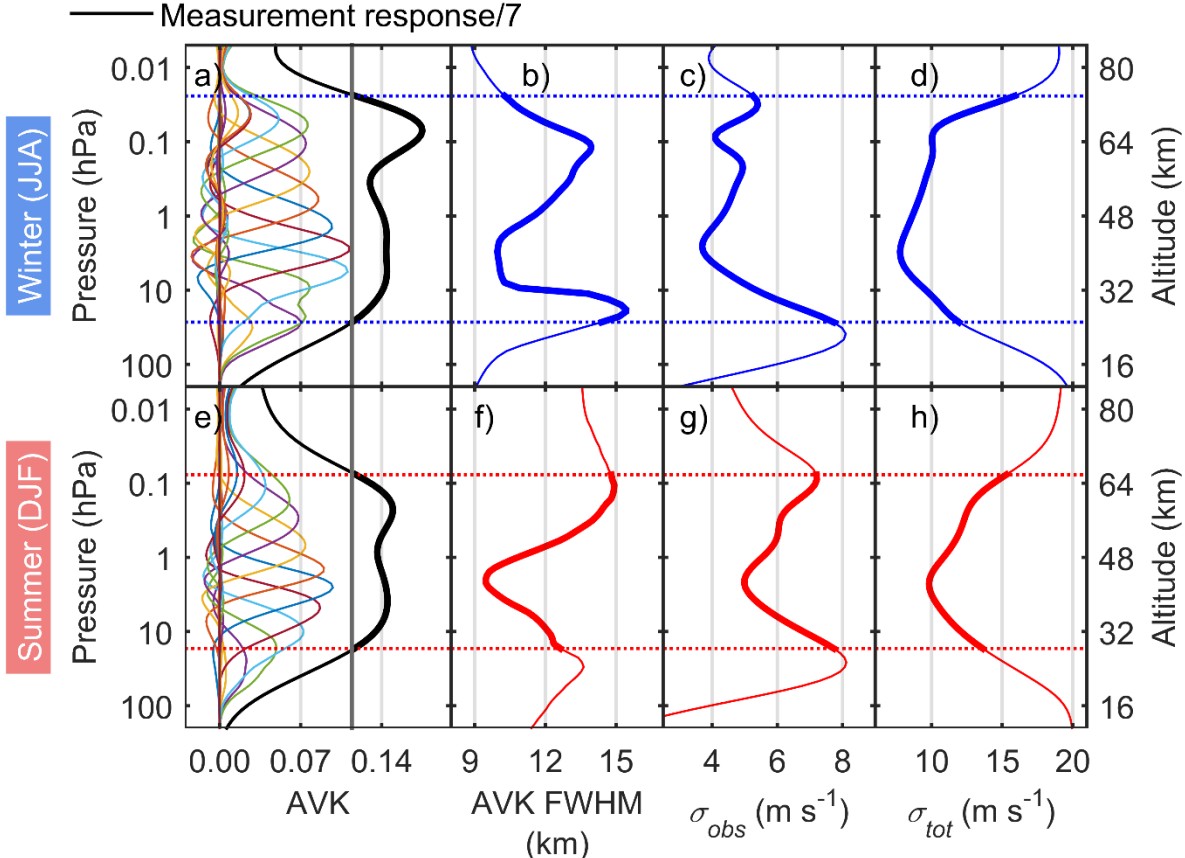

**Figure 6.** Example zonal wind retrievals for simulated clear-sky, 12hr observations (i.e. 6 hrs in east and west directions) of the $O_3$ 231.28 GHz line using a ground-based radiometer with a 1400 K system temperature, 30 kHz frequency resolution, and 60° zenith viewing angle located at Halley station (75°37'S, 26°14'W), Antarctica. In a) and e) every sixth averaging kernel, and the scaled measurement response (MR), are shown for mean winter (JJA) and summer (DJF) conditions respectively. The vertical grey lines in a) and e), dashed horizontal lines, and the thicker sections of the plots indicate where MR ≥ 0.8.





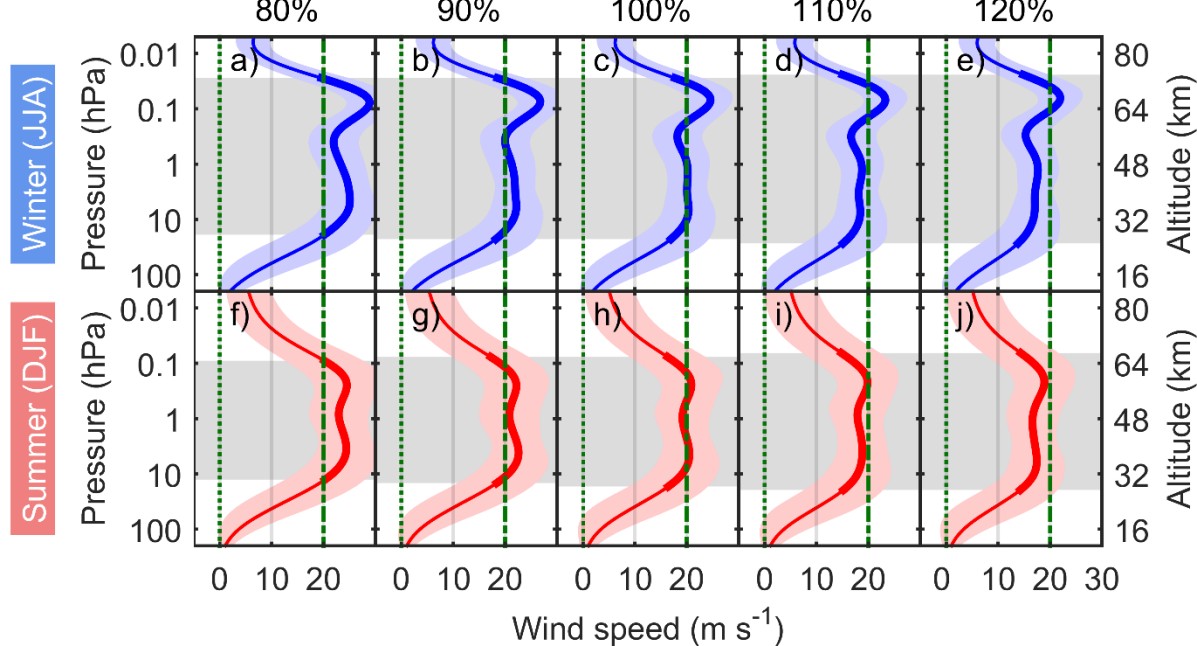

**Figure 7.** Monte Carlo error analyses using 500 repeat zonal wind retrievals for simulated clear-sky, 12hr observations (i.e. 6hrs in east and west directions) of the $O_3$ 231.28 GHz line using a ground-based radiometer with a 1400 K system temperature, 30 kHz frequency resolution, and 60° zenith viewing angle located at Halley station (75°37'S, 26°14'W), Antarctica. The a priori $O_3$ VMR profiles are scaled by 80%, 90%, 100%, 110%, and 120% of the "true" profiles for a) to e), and f) to j) respectively. The mean retrieved winter and summer winds and 1σ errors (shaded areas) are shown, with the thicker sections and the shaded grey panels indicating where MR ≥ 0.8. The true wind profile used to simulate the atmospheric spectra is shown by the dash-dotted green lines. The a priori wind profile is shown by the dotted green lines.



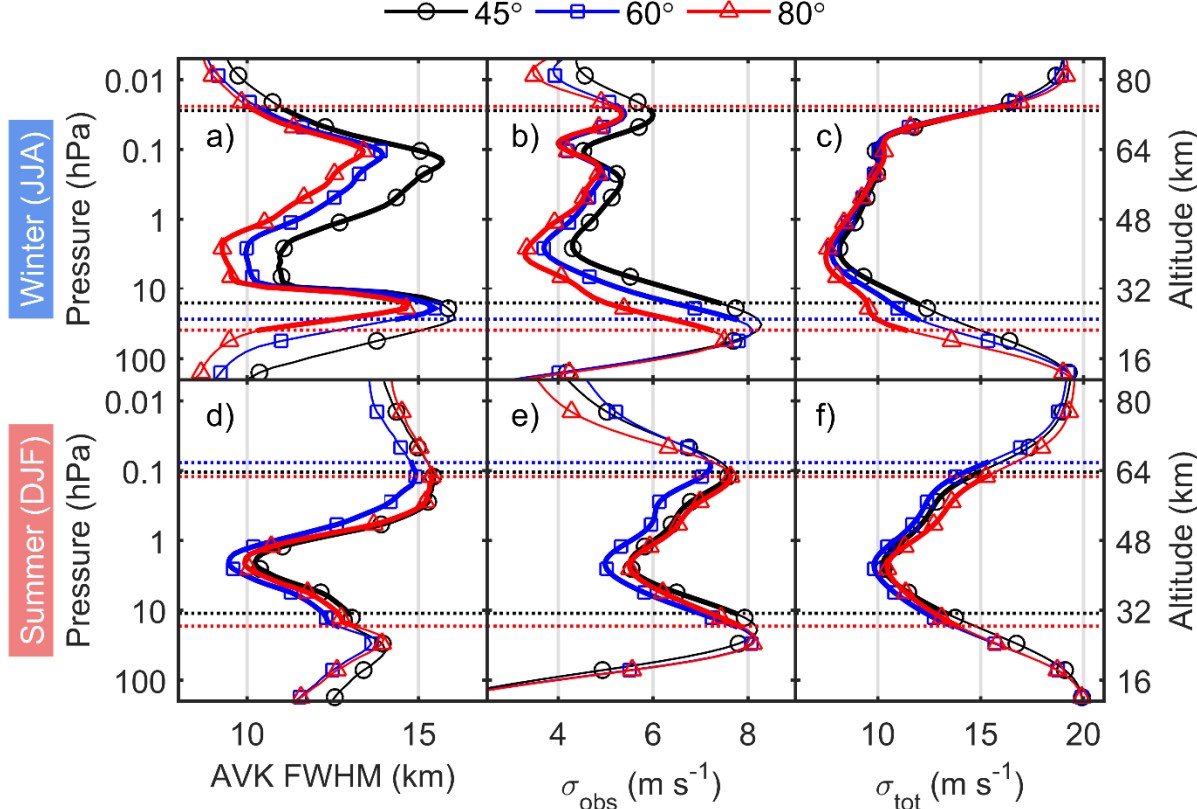

**Figure 8.** Zonal wind retrievals for different zenith viewing angles, where AVK FWHM are the full-width half-maxima of each averaging kernel, $\sigma_{obs}$ is the measurement uncertainty, and $\sigma_{tot}$ is the total uncertainty. Calculations are for simulated clear-sky, 12hr observations (i.e. 3hrs in east and west directions) of the $O_3$ 231.28 GHz line in mean winter (JJA) and summer (DJF) conditions at Halley station (75°37'S, 26°14'W), Antarctica using a ground-based radiometer with a 1400 K system temperature and 30 kHz frequency resolution. The horizontal dashed lines and thicker sections of the curves indicate the pressure / altitude ranges where MR $\geq$ 0.8.





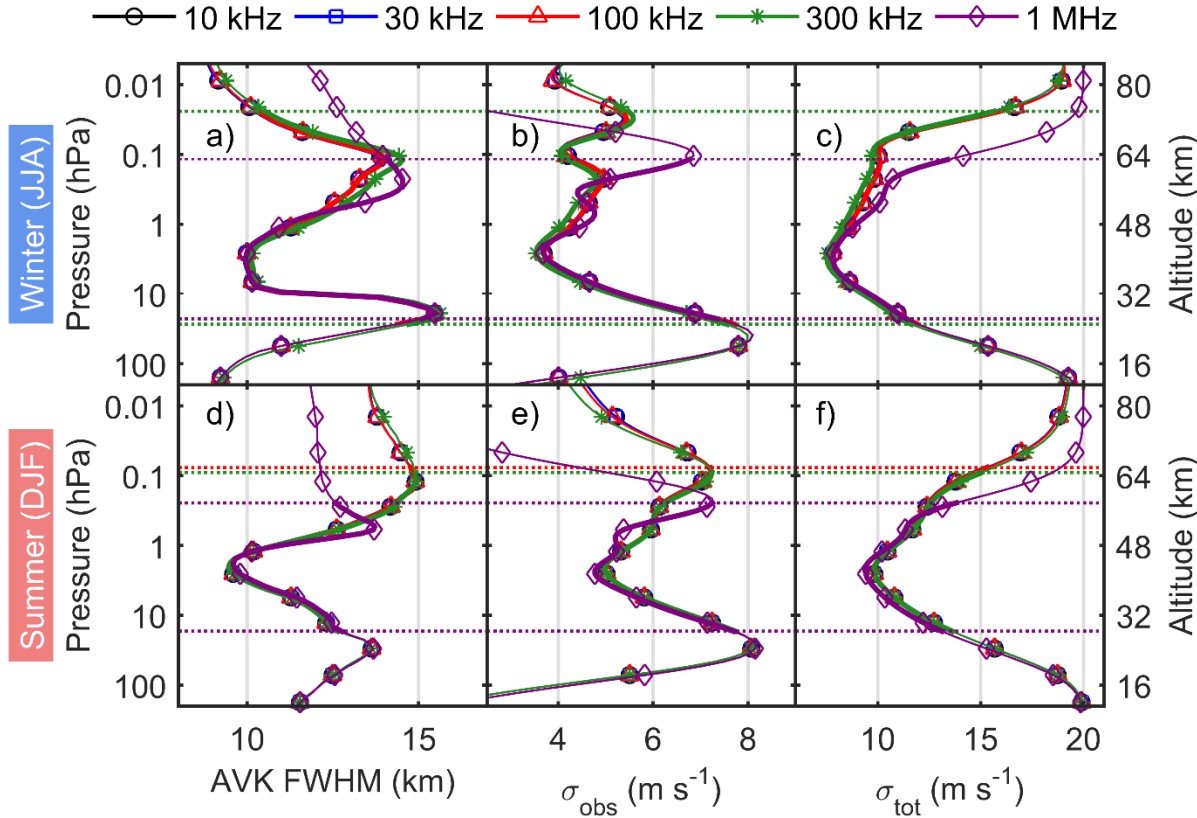

**Figure 9.** As Figure 8, but for different frequency resolutions and a 60° zenith viewing angle.



**Figure 10.** Rows of the Jacobian describing the horizontal wind retrieval, normalised by the layer thickness of the retrieval grid. The data are from zonal wind retrievals for simulated clear-sky, 12hr observations (i.e. 6hrs in east and 6hrs in west directions) of the $O_3$ 231.28 GHz line using a ground-based radiometer with a 1400 K system temperature, 30 kHz frequency resolution, and 60° zenith viewing angle located at Halley station (75°37'S, 26°14'W), Antarctica. The colour scale in a) and b) indicates the values of the Jacobian matrix. Rows of the Jacobian matrix for selected altitude levels are plotted in c) and d). Plots b) and d) show the centre frequencies on an expanded scale.





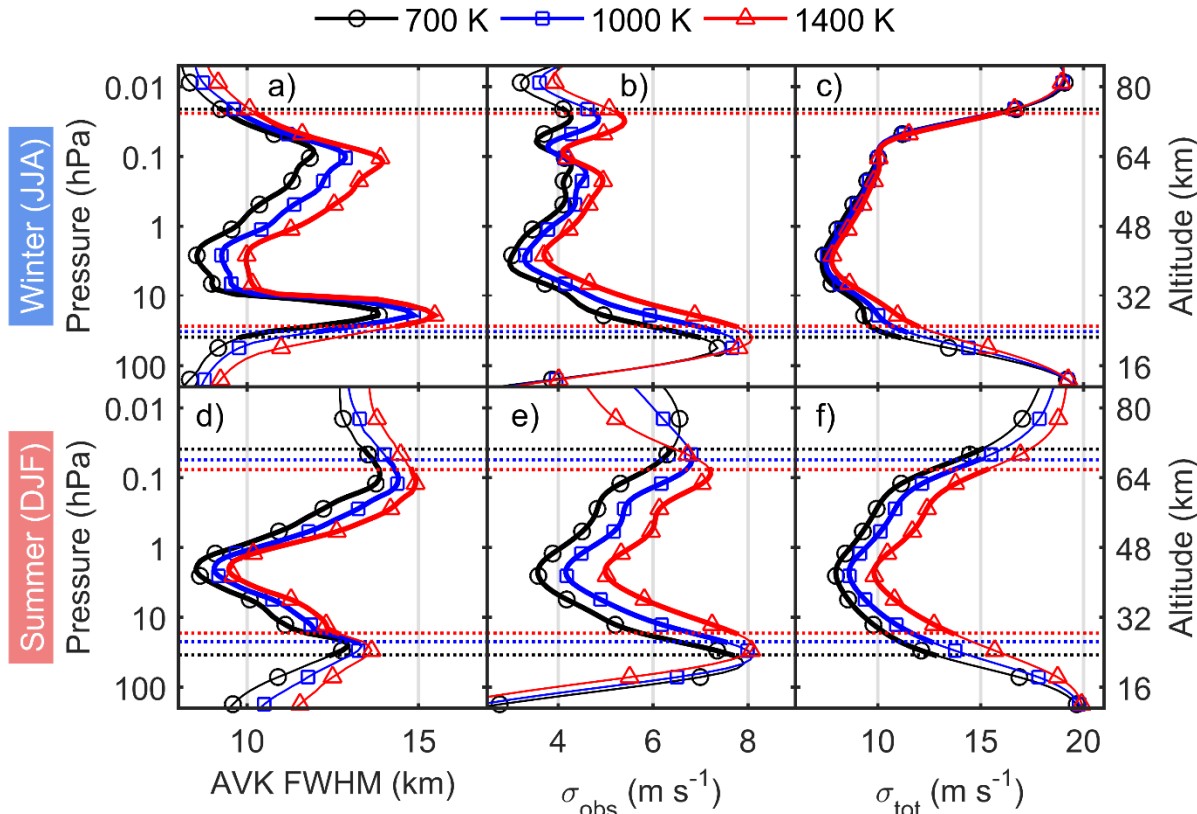

**Figure 11.** As Figure 8, but for different radiometer system temperatures and a 60° zenith viewing angle.



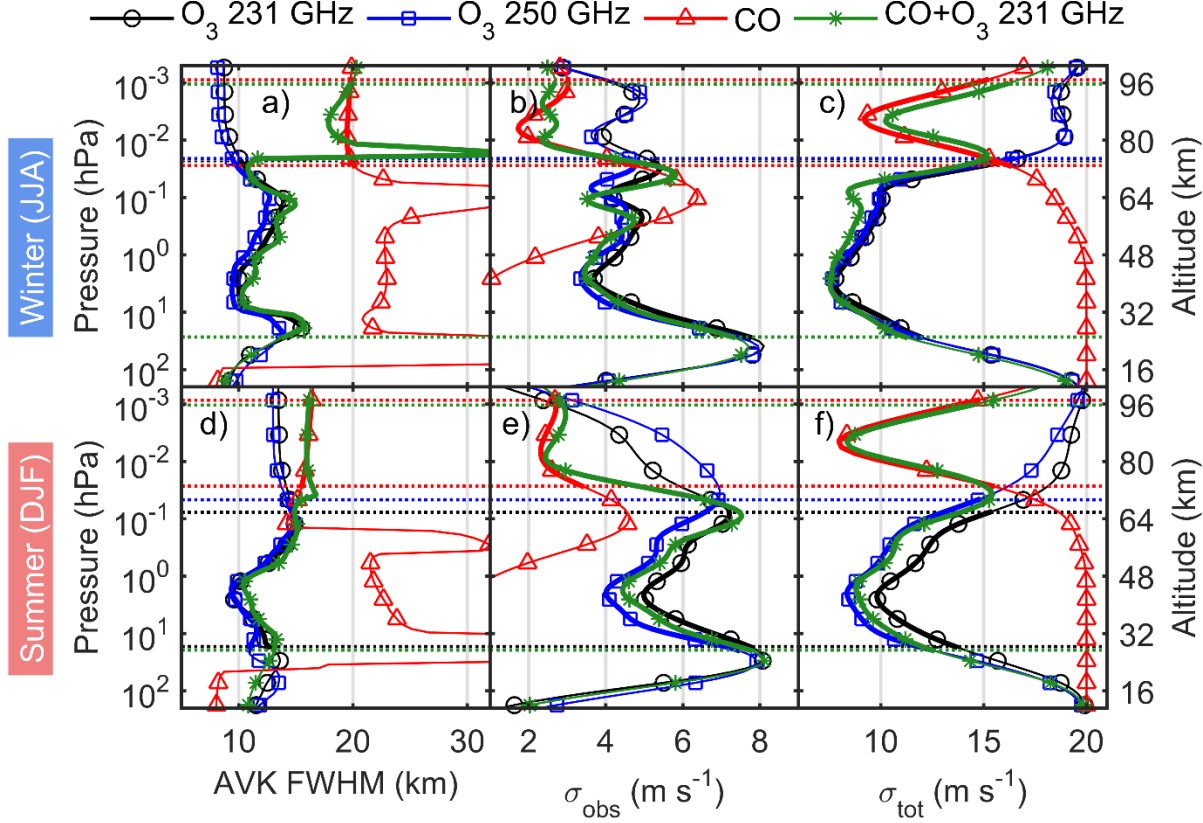

**Figure 12.** As Figure 8, but for retrievals using the $O_3$ 231.28 GHz, $O_3$ 250 GHz, CO 230.54 GHz, and combined CO 230.54 GHz and $O_3$ 231.28 GHz atmospheric lines and a 60° zenith viewing angle.





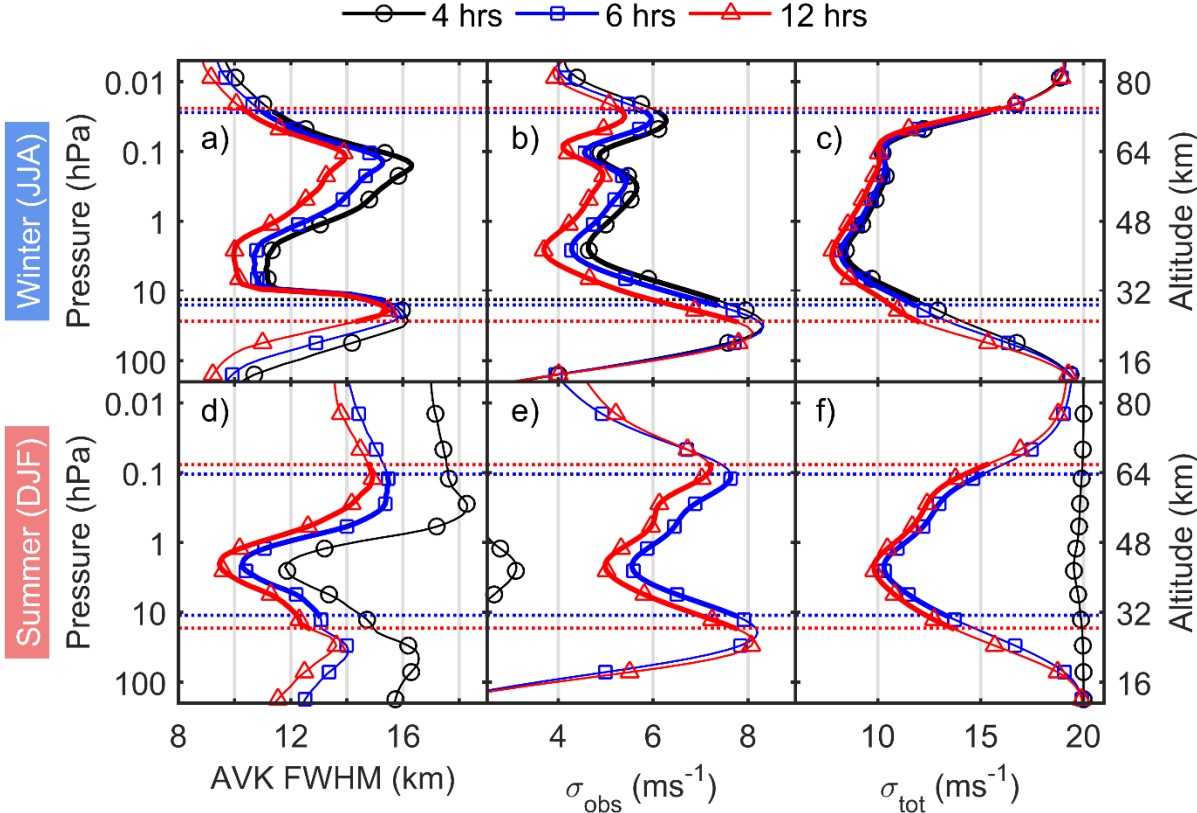

**Figure 13.** As Figure 8, but for different total measurement times and a 60° zenith viewing angle.





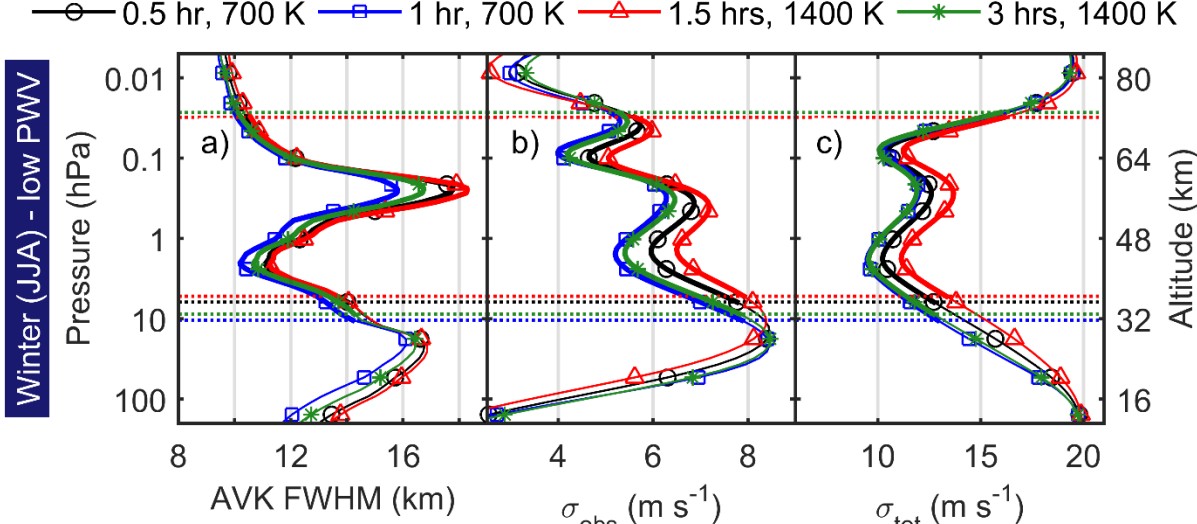

**Figure 14.** As Figure 13, but simulated for different total measurement times and radiometer system temperatures and under optimal winter-time observing conditions at Halley where precipitable water vapour (PWV) is in the lower quartile (below 0.92 mm).