# Peer review of "Measurement of horizontal wind profiles in the polar stratosphere and mesosphere using ground based observations of ozone and carbon monoxide lines in the 230–250 GHz region: Proof of concept"

_Atmospheric Measurement Techniques, 2015_

## Short Comment (SC1) · 5 Feb 2016

Your manuscript is very interesting. It is particularly notable how you benefit from the dry Antarctic atmosphere to push the upper limit of your measurement range to very high altitudes by using CO (in this context it might also be interesting to explain to the reader how you can retrieve at altitudes where Doppler broadening is dominating over pressure broadening). However, on page 4, line 27 I found a statement that disturbs me: You cite our paper (Rüfenacht et al., 2014) of wind retrievals with the wind ra-

diometer WIRA as reference for your statement that baseline issues which can arise from standing waves or other sources are uncritical for wind retrievals from observations with Doppler microwave radiometers. However, it should be noted that in the cited paper we have only analysed the effect of a baseline ripple with a period and amplitude similar the one found in the data acquired with the wind radiometer WIRA. Such a baseline is indeed uncritical. Nevertheless one could imagine that other baselines (e.g. with faster oscillations) can influence wind retrievals. I would be grateful if you could modify your manuscript in this sense.

---

## Author Comment (AC1) · 1 Mar 2016

We thank Dr Rufenacht for his short comment (Interactive comment on Atmos. Meas. Tech. Discuss., doi:10.5194/amt-2015-406, 2016) on our manuscript. We are pleased that he found the manuscript very interesting. Our discussion response, and proposed minor changes to the manuscript to address the two points raised in his short comment, are given below.

[Figure]

1. 'It is particularly notable how you benefit from the dry Antarctic atmosphere to push the upper limit of your measurement range to very high altitudes by using CO (in this context it might also be interesting to explain to the reader how you can retrieve at altitudes where Doppler broadening is dominating over pressure broadening).'

Dr Rufenacht is correct in pointing out that, at the high altitudes where we demonstrate the wind retrievals that would be possible using CO, the linewidth is dominated by Doppler (thermal) broadening rather than pressure broadening. Calculated Doppler- and pressure- broadened full-width half-maxima (FWHM) linewidths for the CO 230 GHz line in mean winter (JJA) and mean summer (DJF) conditions at Halley, Antarctica are shown in Figure 1. Pressure and temperature profiles used in the calculations are taken from the SD-WACCM model data used in the simulated wind retrievals, and the pressure broadening coefficients are from the HITRAN spectroscopic database (http://hitran.org/). The dotted horizontal lines in the figure show the altitudes above which the Doppler contribution to the linewidth exceeds pressure broadening, i.e. above 62 km in winter and above 69 km in summer. Doppler broadening increases rapidly above 100 km as temperature rises in the thermosphere (e.g., see Figure 4e in the manuscript). We retrieve the horizontal wind over a 24 km altitude range between 73 km and 97 km (shown by the green shaded panel in the figure) where the linewidth, dominated by Doppler broadening, is at a minimum.

We propose explaining this in the manuscript by adding the following text on page 11, line 27 after the sentence ending '...ultraviolet photo-dissociation of carbon dioxide (CO2).'

At the retrieval altitudes the CO linewidth is dominated by Doppler (thermal) broadening. However the Doppler FWHM linewidth is at a minimum with a reasonably constant value of 440+/-10 MHz between 70 km and 97 km. Doppler broadening increases above 97 km due to higher temperatures in the lower thermosphere. Pressure broadening dominates, and the CO linewidth rapidly increases, below 62 km in winter and below 69 km in summer. Thus the wind retrieval is possible at high altitudes where

the minimum in the Doppler broadening characterises the altitude and where the CO mixing ratio is sufficiently high, but the height resolution of the retrieval is limited by the uniformity of the Doppler linewidth at these altitudes.

2. 'on page 4, line 27 I found a statement that disturbs me: You cite our paper (Rüfenacht et al., 2014) of wind retrievals with the wind radiometer WIRA as reference for your statement that baseline issues which can arise from standing waves or other sources are uncritical for wind retrievals from observations with Doppler microwave radiometers. However, it should be noted that in the cited paper we have only analysed the effect of a baseline ripple with a period and amplitude similar the one found in the data acquired with the wind radiometer WIRA. Such a baseline is indeed uncritical. Nevertheless one could imagine that other baselines (e.g. with faster oscillations) can influence wind retrievals. I would be grateful if you could modify your manuscript in this sense.'

We accept that the original wording in the manuscript, describing the potential impact of standing waves and other baseline artefacts on millimetre-wave wind retrievals, could be misleading. We propose amending the manuscript to briefly discuss possible strategies for minimising and characterising such baseline effects, by replacing the sentence on page 4, line 27 starting 'The effects of frequency errors arising from reference oscillator instabilities and spectrum baseline artefacts...' as follows.

Rufenacht et al. (2014) showed for the WIRA instrument that frequency errors arising from reference oscillator instabilities and spectrum baseline artefacts such as standing waves are either small or can be adequately characterised to minimise their impact on the wind retrievals. However for other wind radiometers these effects could make a larger contribution to the measurement uncertainty, that is not considered in the simulations here. For example, with instruments using a SIS mixer there is the potential for significant interfering reflections between cryostat windows and other optical components. The potential sources of such artefacts need to be identified at the instrument design and build stages and steps taken to reduce them to an acceptable level, e.g.

through anti-reflection machining of optical surfaces and path-length modulation aimed at minimising standing wave amplitudes.

[Figure]

Fig. 1 shows a plot with Altitude (km) on the y-axis (0 to 120) and FWHM linewidth (kHz) on the x-axis (300 to 900).

CO 230 GHz:
Wind retrieval altitude
range (73-97 km)

69 km

62 km

**Pressure**
—— CO summer
—— CO winter
**Doppler**
- - - CO summer
- - - CO winter

**Fig. 1.** Pressure (air) and Doppler broadened full width half maximum (FWHM) linewidths for the CO 230.54 GHz emission line above Halley station, Antarctica in summer (DJF) and winter (JJA).

---

## Referee Comment (RC1) · Anonymous Referee #1 · 7 Mar 2016

Ground-based measurements of winds in the stratosphere and mesosphere are sparse, basically due to a lack of a relative simple instrument to make such observations. Based on older efforts, some quite recent studies have put emphasis on that microwave radiometry can derive wind information, both from satellite (Baron et al., 2013) and ground (Rüfenacht et al., 2012 and 2014). Rüfenacht et al. selected an ozone transition at 142 GHz, while in this manuscript measurements using some transitions around 230 GHz are explored by simulations.

[Figure]

The transition selected by Rüfenacht et al. is probably the best general choice for ground-based observations, while transitions above 200 GHz can be of interest for high altitude and polar sites. The manuscript describes very clearly and in a detailed manner the potential of using the 230 GHz transitions for conditions found above Antarctica. Based on the experience from the articles mentioned above, it is no surprise that winds can be retrieved also from these transitions, but it is still a valuable contribution to point out and document these options. The manuscript is very well written and this manuscript should be published in AMT after some minor corrections and consideration of the title.

The text is concisely written, but the number of figures still makes the manuscript quite long and gives an impression of a technical report. I think some figures can be removed. A figure to explain the effective area at different altitudes and elevations should not be needed and Figure 2 could be removed. Figure 4 is neither critical, it suffices to comment the important features in the text.

I am also a bit sceptic about Fig 9. First of all, it could suffice to mention in the text that there is no change in the performance for resolutions up to 300 kHz. However, this figure also brings up the main drawback of the manuscript, it just includes simulations assuming a more or less ideal instrument and it can be questioned if this is a "proof of concept" (more below). My point here is that a perfect knowledge of the "instrumental line shape" is assumed and then very good results can be obtained in simulations, even for relatively poor resolutions, but these results are necessarily not valid for practical measurements. The impact of errors in the assumed instrumental line shape should increase strongly when deteriorating the frequency resolution, and in practice a high frequency resolution is always to prefer. That is, for me Fig 9 is of little interest, it could even be misleading. The authors should also consider if the assumption of an ideal instrument affects other results. For example, could antenna sidelobes be a problem for a 80 degree zenith angle?

My second main comment is the assumption of a constant wind field over the areas

shown in Fig 2. The spatial resolution is well described on lines 10-16 of page 5. When the text then jumps to ECMWF data, I then expected a discussion of gradients inside the coverage area, but instead just wind speed histograms are discussed. That is, at least a new paragraph should be started on line 19.

Even better would be if wind gradients could be also discussed. A hint to the authors is to include wind gradients in the Monte Carlo simulations. I don't demand this extension, but it would rise the value of the manuscript considerably, as the manuscript would introduce some really new analysis (as far as I know).

Two smaller comments:

Page 3, line 25: Is not the main frequency uncertainty normally originating from the mixer LO signal?

Page 10, lines 24-26: No information is lost by switching to a higher frequency resolution. The lowed "SNR" in individual channels is compensated by the higher number of channels. That is, I don't agree with the argumentation here.

Don't the term "proof of concept" imply a practical demonstration, to show that an idea or some simulated results work in practice? This was at least my understanding of the term before reading this manuscript, and I think this description from Wiktionary supports this view:

A short and/or incomplete realization of a certain method or idea to demonstrate its feasibility.

That is, please, consider changing the title.

---

## Referee Comment (RC2) · Anonymous Referee #2 · 1 Apr 2016

The paper by Newnham et al. presents a sensitivity study that shows the potential of ground-based radiometers for measuring high latitude middle atmospheric winds. The effects of the main observational parameters are discussed. This paper addresses an important topic since very few systems provide such information on a routinely basis though radiometers are widely used for trace gas measurements. I found the paper very clear and it should be published in AMT. I have, however few comments.

General comments:

[Figure]

1) The diurnal variation of O3 is not discussed in the paper. I believe it can significantly impact the wind measurement performances above 60 km. It would be interested if the difference between day and night measurement performances could be assessed.

2) I don't believe that the retrieval calculations performed by the authors fully describe the instrument potential. The altitude dependent retrieval vertical resolution is given by the fixed "retrieval parametrization" including a priori information and probabilistic optimization. I see two problems.

A) The method brings unnecessary subjectivities in the results: results would have been different if a different wind a priori uncertainty is used. (the justification that uncertainties are realistics is not satisfactory since they change with time, altitude, location and a priori data).

B) The best solution depends on the scientific target of the instrument and the vertical resolution is as important as the precision. So, I would have been interested to see error budgets for fixed vertical resolutions (8 km, 10 km, ...). It is then possible to define both the vertical resolution and the observation time to obtain the most satisfactory products. This will be translated into constraints on the a priori parameters for processing the real data (but this is outside the scope of this manuscript).

Such error budget with no a priori contamination and fixed vertical resolution can be assessed, even if the OEM formalism is used. However my point is not to ask the authors to change the study since their approach is commonly used and the results are not wrong. I am simply interested to hear the authors' comment on my comment.

Specific comments:

P3, line 24: The sentence is ambiguous: zonal and meridional winds are derived from perpendicular azimuthal directions. Measurement biases on each component are removed from measurement at opposite directions.

P3, line 30: "showed good agreement . . . 10%". This positive statement seems to be in
contradiction with that in Line 2 which sounds negative: "deviate increasingly above 40 km ...STD exceeds 20 m/s". Should we consider that measurements and (re-)analysis are in good agreements in the stratosphere?

P4, line 21: Why such a large difference between the receiver temperature (200K) and the system temperature (1400K)? Could we expect a smaller difference when designing a new instrument?

P4, line 28: I agree with the comment published by R. Rufenacht in the discussion. I think the instrument in Rufenacht et al. (2014) has been carefully designed to mitigate standing waves which, otherwise, could be one of the most significant source of errors. The sentence should be rephrased.

P6, lines 1-3: Are day and night profiles averaged all together? The O3 diurnal variation has an effect on the wind measurement precision and altitude coverage above 60 km.

P7, line 15: Is the covariance matrix set with respect to the horizontal wind or to the line-of-sight wind?

P8, Sec. 2.3: It is not clear for me whether the error estimations are for the average of the two retrievals with opposite directions or a single direction retrieval.

P9, lines4-5: I agree that for real wind retrievals, the O3 priori information is good enough to obtain small uncertainties due to the O3 a priori. However, in my point of view, the calculated errors are too large. The wind retrievals should be rather independent of the O3 a priori? May-be there is an additional error induced by the retrieval procedure itself and, in that case, the other retrieval errors estimated in the paper might also be overestimated. a) Authors should check if the O3 a priori error estimated with the MC analysis match the errors linearly derived from the off-diagonal terms of the averaging kernel. b) why the average of two retrievals with opposite directions does not remove such error?

P10 and 11: Both sections 3.3 and 3.4 deal with the dependence of the retrieval performances with respect to the signal to noise ratio: increasing the observation time by a factor 4 is similar to reducing the system temperature by a factor 2. I think the discussion should be for both the system temperature and the observation time as in Fig 14. So I would recommend to merge the two sections as well as Fig.11 and 13. (Note that the figure numbers is the text is not consistent with their actual numbers).

P31,32,33: Fig.12 should be Fig.14, Fig.13 should be Fig.12 and Fig.14 should be Fig.13 ?

P27, Fig8 caption: Should be 6hours in each opposite direction (value in parenthesis)?